# *Staphylococcus schweitzeri*—An Emerging One Health Pathogen?

**DOI:** 10.3390/microorganisms10040770

**Published:** 2022-04-02

**Authors:** Chantal Akoua-Koffi, Adèle Kacou N’Douba, Joseph Allico Djaman, Mathias Herrmann, Frieder Schaumburg, Silke Niemann

**Affiliations:** 1Centre Hospitalier Universitaire de Bouaké, Bouaké P.O. Box BP 1174, Côte d’Ivoire; cakoua26@gmail.com; 2Department of Biology, Université Alassane Ouattara de Bouaké, Bouaké P.O. Box BP V18, Côte d’Ivoire; 3Training and Research Unit of Medical Sciences, Félix Houphouët-Boigny University, Abidjan P.O. Box BP 44, Côte d’Ivoire; knadele@yahoo.fr; 4Centre Hospitalier Universitaire Angré, Abidjan P.O. Box BP 1530, Côte d’Ivoire; 5Training and Research Unit of Biosciences, Félix Houphouët Boigny University, Abidjan P.O. Box BP V 34, Côte d’Ivoire; djamanj@yahoo.fr; 6Institute of Medical Microbiology, University Hospital Münster, 48149 Münster, Germany; mathias.herrmann@uni-muenster.de (M.H.); frieder.schaumburg@ukmuenster.de (F.S.)

**Keywords:** *Staphylococcus schweitzeri*, zoonosis, Africa, One Health

## Abstract

The *Staphylococcus aureus*-related complex is formed by the *Staphylococcus aureus*, *Staphylococcus schweitzeri*, *Staphylococcus argenteus*, *Staphylococcus roterodami* and *Staphylococcus singaporensis*. Within this complex, *S. schweitzeri* is the only species mainly found in African wildlife, but it is rarely detected as a colonizer in humans or as a contaminant of fomites. The few detections in humans are most likely spillover events after contact with wildlife. However, since *S. schweitzeri* can be misidentified as *S. aureus* using culture-based routine techniques, it is likely that *S. schweitzeri* is under-reported in humans. The low number of isolates in humans, though, is consistent with the fact that the pathogen has typical animal adaptation characteristics (e.g., growth kinetics, lack of immune evasion cluster and antimicrobial resistance); however, evidence from selected in vitro assays (e.g., host cell invasion, cell activation, cytotoxicity) indicate that *S. schweitzeri* might be as virulent as *S. aureus*. In this case, contact with animals colonized with *S. schweitzeri* could constitute a risk for zoonotic infections. With respect to antimicrobial resistance, all described isolates were found to be susceptible to all antibiotics tested, and so far no data on the development of spontaneous resistance or the acquisition of resistance genes such the *mecA*/*mecC* cassette are available. In summary, general knowledge about this pathogen, specifically on the potential threat it may incur to human and animal health, is still very poor. In this review article, we compile the present state of scientific research, and identify the knowledge gaps that need to be filled in order to reliably assess *S. schweitzeri* as an organism with global One Health implications.

## 1. Introduction

The One Health concept recognizes that the health of humans is closely connected to the health of animals and the environment. Among bacterial pathogens, staphylococci have proven to be appropriate models for “One Health” studies, as certain species and clones can be transmitted from humans to animals (wildlife and farm animals) and vice versa [1]. *S. aureus* causes high morbidity and high attributable mortality, and is therefore a major threat to public health and an economic burden on health systems worldwide. Infection with *S. aureus* also poses a major health risk in animals (e.g., bovine mastitis) [2]. Wild animals are potential reservoirs for the transmission of *S. aureus*. Frequent contact between wild animals and livestock as well as contact with humans increase the transmission of bacteria and enhance the risk of colonization and infection in humans and animals [1].

A few years ago, whole genome sequencing (WGS) revealed that isolates from several specific *S. aureus* lineages are divergent from classical *S. aureus*, and they were designated as closely related but separate coagulase-positive species. They have been assigned as *S. argenteus* and *S. schweitzeri* [3,4]. Together with *S. aureus*, these species form the *S. aureus* complex. Only recently, two more members of the *S. aureus* complex have been described: *S. roterodami* and *S. singaporensis* [5,6]. While *S. aureus*, *S. argenteus*, *S. roterodami* and *S. singaporensis* have been isolated from human infections, *S. schweitzeri* has been found almost exclusively as a colonizer of fruit bats and non-human primates in West and Central Africa, but not as a colonizer or cause of infection in humans [3,5,6,7,8,9].

*S. schweitzeri* is named after Albert Schweitzer, Nobel Peace Prize Laureate and founder of the Hôpital Albert Schweitzer in Lambaréné (in formerly French Equatorial Africa, today Gabon) [4].

The bacterium can easily be mistaken for another species of the *S. aureus* complex using culture-based routine techniques. For this reason, colonization or even infection in humans or domestic animals by *S. schweitzeri* might have gone unreported. Since it produces pathogenic factors such as enterotoxins, toxic shock syndrome toxins, exfoliative toxins, hemolysins, adhesins and autolysins [3,8,9,10], it is conceivable that *S. schweitzeri* has the ability to cause infections in humans.

Given the possible transmission of *S. schweitzeri* from wildlife to humans or livestock, and the possible adaptation of this bacterium to humans, it is important to understand the physiology of this potential One Health pathogen and to develop conventional identification methods to better assess its prevalence. This review aims to summarize existing information on *S. schweitzeri* and to identify knowledge gaps.

## 2. The *S. aureus* Complex

*S. aureus, S. schweitzeri, S. argenteus, S. roterodami* and *S. singarporensis* form the *S. aureus* complex [3,4,5,6]. The members of the *S. aureus* complex other than *S. schweitzeri* will be briefly introduced here.

*S. aureus* is a globally occurring human commensal, as well as a pathogen in humans and animals [11,12,13]. Infections caused by *S. aureus* are of particular concern due the virulence of the pathogen, as well as to the frequent occurrence of antimicrobial (multi) resistance [14]. Given the wealth of literature on *S. aureus*, any more detailed discussion here is precluded for the sake of the review format of this article.

*S. argenteus* was originally described as a community-acquired (CA) methicillin-resistant *S. aureus* (MRSA) lineage belonging to the clonal complex (CC) CC75. The first isolates were recovered from patients with CA-MRSA infections in Australian Aboriginal communities [15]. Subsequently, the complex was delineated as *S. argenteus* [4]. In the meantime, *S. argenteus* has been found worldwide, e.g., in Thailand, Japan, France, Belgium, Trinidad, Tobago and Africa, both in humans and animals; however this species seems to be predominantly associated with humans [3,8,16,17,18,19,20,21,22,23]. *S. argenteus* and *S. aureus* differ in the amino acid sequence contained in peptidoglycan, while *S. argenteus* and *S. schweitzeri* possess the same peptidoglycan type [4]. At the molecular level, the nucleotide sequences of *S. argenteus* display 87.4% identity with *S. aureus* [24]. The pathogenicity of *S. argenteus* appears to be comparable to that of *S. aureus*, and 76.6% of the virulence genes of *S. aureus* have also been found in *S. argenteus* [24]. Some isolates also produce Panton–Valentine leukocidin (PVL) [15,25], a toxin associated with *S. aureus* isolates causing skin and soft tissue infections [26,27]. The absence of the pigment staphyloxanthin in *S. argenteus* results in the whitish color of the colonies [4,28]. Many *S. argenteus* isolates are penicillin-resistant [17], while other antibiotic resistances are less frequent; for instance, methicillin-resistance was found in isolates from Australia or Denmark [15,29].

*S. roterodami* was first described in 2021 by Schutte et al. [6]. This species was isolated from a human foot wound infection in the Netherlands. However, the patient probably became infected with *S. roterodami* in Bali after suffering an injury to his foot. To our knowledge, there is only this one isolate described so far. The strain was originally identified as *S. argenteus*. Antimicrobial susceptibility was determined using the Vitek 2 susceptibility testing card for Gram-positive bacteria, and *S. roterodami* was found to be susceptible to all antibiotics tested. An analysis of cellular fatty acids showed several differences in fatty acid proportions compared to *S. aureus*, *S. argenteus* and *S. schweitzeri* [6].

In 2021, *S. singaporensis* was described by Chew et al. following a retrospective clinical laboratory cohort study of non-*S. aureus* complex members in Singapore [5]. Six out of forty-three isolates stood out as genetically distinct from *S. argenteus*. *S. singaporensis* was found in infectious diseases in humans, and the clinical features of these infections were similar to *S. aureus* infections. The isolates did not contain some of the virulence genes typically associated with *S. aureus*, i.e., staphylococcal enterotoxins, *tst* (toxic shock syndrome Toxin-1), or *pvl* (Panton–Valentine leukocidin). The *S. singaporensis* isolates did not reveal antibiotic resistance [5].

## 3. Prevalence of *S. schweitzeri*

To date, *S. schweitzeri* isolates have only been reported from sub-Saharan Africa, namely Gabon, Nigeria, DR Congo and Côte d’Ivoire (Table 1). The isolates were obtained almost exclusively from animals, fruit bats, non-human primates and a gorilla. In three cases, colonization in humans was detected, but these were probably spillover events from handling animals that carried *S. schweitzeri* [3,10,30]. This low rate was actually surprising, because *S. schweitzeri* has been found in animals that serve as bushmeat [9]. Infections caused by *S. schweitzeri* have not yet been reported. Livestock colonized or infected with *S. schweitzeri*, such as goats or cattle, have also not yet been found in a large multi-center study [9].

**Table 1 microorganisms-10-00770-t001:** Prevalence of *Staphylococcus schweitzeri* in humans and animals.

Source	Prevalence (n/Total n (%))	Country	Year	Setting	Antimicrobial Resistance	Ref.
Fomites	2/239 (0.8%)	Nigeria	2015–2016	Contamination	None detected	[31]
Fruit bat *(Eidolon helvum)*	11/250 (4.4%)	Nigeria	2015–2016	Fecal colonization	None detected	[8]
Fruit bat *(Rousettus aegyptiacus)*	2/55 (4%)	Gabon	2015	Colonization	None detected	[32]
Human	1/500 (0.2%)	Gabon	2009	Colonization	None detected	[33]
Human	2/1014 (0.2%)	Gabon	2012–2013	Colonization	Not stated	[30]
Monkeys	1/12 (8%)	Côte d’Ivoire	2012	Colonization	None detected	[9]
Monkeys	17/71 (24%)	Gabon	2010–2013	Colonization	None detected	[9]
Monkeys	6/10 (60%)	DR Congo	2011	Colonization	None detected	[9]
Wildlife	0/2855 (0%)	Austria, Germany, Sweden	Not stated	Colonization and infection	Not applicable since no *S. schweitzeri* was found	[34]

Important information: *S. schweitzeri* isolates originally identified as *S. aureus* by Ngoa et al., 2012 [33] are also included in this table.

## 4. Identification

Identification by routine methods (e.g., catalase, coagulase, MALDI-TOF) is challenging, as these diagnostic approaches cannot yet delineate the species of the *S. aureus*-related complex reliably, and it is likely that *S. schweitzeri* remains unnoticed in routine settings, particularly in low-resource settings where genomic surveillance is not in place. Although *S. schweitzeri* is included in MALDI-TOF databases (e.g., MBT Compass IVD software, version 4.2.90, Bruker, Bremen), the current profiles do not reliably distinguish between *S. schweitzeri* and other members of the *S. aureus*-related complex (this is our own observation). Suspicious signs that point towards *S. schweitzeri* are: (i) isolates detected in animals (bats, monkeys) or humans with close contact to these animals; (ii) double hemolytic zones on blood agar (Figure 1); and (iii) the absence of any antimicrobial resistance. However, definite species confirmation requires genetic analyses.

**Figure 1 microorganisms-10-00770-f001:**
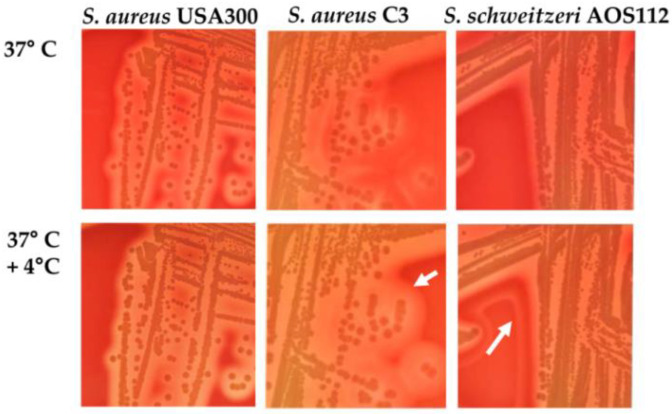
Hemolysis of *Staphylococcus aureus* USA300, *S. aureus* C3, and *Staphylococcus schweitzeri* AOS112. Cultivation on Columbia blood agar at 37 °C overnight, followed by cooling to 4 °C for 24 h results in an enhanced hemolysis pattern plates only with *S. aureus* C3 (animal adapted goat isolate with intact *hlb* gene [35]) and *S. schweitzeri* AOS112 [31] (arrows indicate double hemolysis) but not with the human-adapted *S. aureus* USA300 [36] (digital photos for this review article were taken after incubation at 37 °C and after further incubation at 4 °C).

### 4.1. PCR

A thermostable nuclease (Nuc1) is part of the core genome of *S. aureus* (i.e., present in almost all *S. aureus* isolates). *S. schweitzeri* possess a divergent thermostable nuclease, and the encoding gene (*nucM*) cannot be amplified by the primers proposed by Brakstad et al. [37] for the confirmation of *S. aureus* due to mismatches (2 of 21 nucleotides and 5 of 24 nucleotides) at the two primer binding sites [37]. Thus, using the PCR by Brakstad et al., *S. schweitzeri* is *nuc*-negative, while *S. aureus* and *S. argenteus* are *nuc*-positive.

In addition, Zhang et al. proposed a PCR targeting the nonribosomal peptide synthetase (NRPS) gene to distinguish *S. aureus* from *S. schweitzeri* and *S. argenteus* based on the length of the amplificate (*S. aureus*: 160 bp; *S. schweitzeri* and *S. argenteus*: 340 bp) [38].

While the results of *nuc* and NRPS PCR already allow a good differentiation between *S. schweitzeri* and *S. argenteus* (and *S. aureus*), high similarities are a reason for confusion with *S. roterodami*/*S. singaporensis* species, and necessitate sequencing methods for definitive identification (Appendix A).

### 4.2. Sequencing

Several sequence-based genotyping methods are available to confirm the species of *S. schweitzeri*. Although *spa* typing gives a hint towards *S. schweitzeri*, this method can be ambiguous even in *spa* types typical of *S. schweitzeri* because *spa* typing and multilocus sequence typing are discordant in 10% of isolates (assessed in an American collection of *S. aureus*) [39].

Recent studies applied multilocus sequence typing (MLST) or whole genome sequencing to distinguish the members of the *S. aureus*-related complex [5,6,10,31]. Phylogenetic trees are constructed to identify the closest related species within the *S. aureus* complex. This implies that the respective candidate species are included in the queried database. For correct species identification, a standard set of type strains of *S. schweitzeri* (FSA084^T^ = DSM 28300^T^ = SSI 89.004^T^), *S. argenteus* (MSHR1132^T^ = DSM 28299^T^ = SSI 89.005^T^), *S. roterodami* (DSM111914^T^ = JCM34415^T^), *S. singaporensis* (SS21^T^ = DSM 111408^T^ = NCTC14419^T^) and *S. aureus* (DSM20231^T^ = ATCC12600^T^) should be included in any phylogenetic tree for species delineation [4,5,6].

The concatenated sequences of the alleles of the seven housekeeping genes in the *S. aureus* MLST typing scheme reliably delineate species-specific clusters in a Neighbor-Joining tree (Figure 2). Such a delineation in different species-related clades holds true for other phylogenetic methods (e.g., maximum likelihood method, Appendix A). Similarly, the core genome MLST (cgMLST) for *S. aureus* including 1861 targets can be used to construct phylogenetic trees with a higher resolution [10].

Sequencing of the 16s RNA gene can be used to delineate *S. schweitzeri* from other members of the *S. aureus*-related complex, despite only a few polymorphisms (Figure 3) [40]. However, discriminatory power might not be sufficient if only partial sequences of the 16s RNA gene are used. The sequencing of *nuc* was suggested by Schutte et al. to discriminate *S. roterodami* from *S. schweitzeri* [6]. However, due to the variance of *nuc* in *S. schweitzeri*, this approach appears inappropriate as *S. roterodami* and *S. schweitzeri* are located in the same cluster (Figure 3a). Whether a correct discrimination of the members of the *S. aureus* complex is possible via sequencing of the 16s RNA gene or *nuc* still needs to be investigated in more detail.

Therefore, to reliably identify *S. schweitzeri*, we suggest two screening steps by PCR (*nuc* and NRPS) and a confirmation either by MLST or WGS (Figure 4).

## 5. Antibiotic Resistance

All *S. schweitzeri* isolates found so far were susceptible to all antimicrobial agents tested, and no methicillin-resistant isolates have been reported yet (Table 1). Only in one study was no information given regarding the antimicrobial resistance of *S. schweitzeri*. [30]. Antimicrobial susceptibility testing was conducted in almost all studies using Vitek2 automated test systems (bioMérieux, Marcy l’Etoile, France), and by applying European Committee on Antimicrobial Susceptibility Testing (EUCAST) clinical breakpoints [8,9,31,32]. In one study, isolates were subjected to antimicrobial susceptibility testing using the disc diffusion method according to Clinical and Laboratory Standards Institute (CLSI) [33]. Whether *S. schweitzeri* is able to produce beta-lactamases on exposure to beta-lactam antibiotics, or whether it can even become methicillin-resistant by acquiring the staphylococcal cassette chromosome *mecA*/*mecC* from *S. aureus* is not known and still needs to be elucidated.

## 6. Pathogenicity

Although no clinical infections with *S. schweitzeri* have been detected yet, it has been shown in vitro that *S. schweitzeri* has many of the pathogenicity factors already known from *S. aureus*. Of 111 virulence genes of *S. aureus* examined, 86 (77.5%) were also found in *S. schweitzeri*. It has been discussed that the pan-genome of these bacteria contains all virulence genes necessary for the pathogenicity of *S. aureus* [24].

*S. schweitzeri* belongs to the coagulase-positive staphylococci and most isolates are able to coagulate human, rabbit, canine and equine plasma, but not porcine or avian plasma [10,41]. Capsular polysaccharides (CP) are virulence factors that can protect the pathogen to evade opsonophagocytic killing [42]. Many *S. schweitzeri* isolates contain the *cap5* gene encoding for CP type 5, and only few contain genes encoding for CP type 8 [10,24]. In contrast, African *S. aureus* isolates from humans as well as from non-human primates were shown to contain the genes for CP5 and CP8, either equally distributed (non-human primates) or biased towards CP8 (humans) [43,44,45]. Many *S. schweitzeri* isolates are positive for the *edinB* (epidermal cell differentiation inhibitor) gene in combination with the exfoliative toxin D (*etd*) [10]. EDIN catalyze the inactivation of RhoA, a regulator of the host cell actin cytoskeleton, while ETD is a serine protease [46,47]. Other protease genes found in the genome of *S. schweitzeri* are *sspA* and *sspP,* but not *aur* or *sspB* [10].

The pathogen is capable of forming a biofilm in vitro [10]. Numerous virulence factors regulate biofilm formation such as the thermostable nuclease Nuc. The primary structures of Nuc from *S. schweitzeri* (NucM) and *S. aureus* (Nuc1) were found to be different (identity 78.1–80.4%, similarity 92.4–94.1%, Figure 3); however, the nuclease activities were identical, as well as the biofilm formation [10,40]. In experiments, the ability of *S. schweitzeri* to form biofilms was significantly lower than that of *S. epidermidis*, which may be related to the fact that the genome of *S. schweitzeri* harbors the genes *icaC* and *icaD*, which are important for biofilm formation, but neither *bap* nor *icaA* [10].

The *S. schweitzeri* genome can contain enterotoxin genes such as *seb*, *seg*, *seh, sei, sel, sen* and the toxic shock toxin (*tst*) gene. For alpha-toxin (Hla), a membrane-damaging toxin that can lead to cell permeability and induce cell necrosis, higher released Hla protein levels were detected in *S. schweitzeri* than in *S. aureus* [48]. We demonstrated a cell-destructive efficacy of *S. schweitzeri* supernatant on A549 cells, a human alveolar epithelial cell line, and on Vero cells, a kidney epithelial cell line derived from an African green monkey, comparable to that of *S. aureus* [10]. This cytotoxic effect might also be attributable to alpha-toxin.

Traditionally, *S. aureus* has been considered a pyogenic extracellular pathogen, but *S. aureus* can also be taken up into host cells, such as epithelial and endothelial cells [36]. In this process, *S. aureus* binds to the α5β1-integrin of the host cell via a fibronectin bridge with the help of the fibronectin-binding proteins (FnBPs). This leads to integrin clustering, the initiation of intracellular signaling cascades, and finally reorganization of the actin cytoskeleton and *S. aureus* uptake into the host cell [49]. The adhesins FnBPA and FnBPB, which are important for the binding of *S. aureus* to fibronectin and thus also for uptake into host cells, are also present in *S. schweitzeri*. In line with this, *S. schweitzeri* can be taken up by epithelial cells in the same way as *S. aureus* [10]. The invasion of non-professional phagocytes is followed by the activation of the host cells, leading e.g., to an increase in cytokine production. In addition, many *S. aureus* isolates are able to translocate from the phagolysosomes into the cytoplasm, similarly to *S. schweitzeri* [10,50]. This phagolysosomal escape is a prerequisite for the bacteria to kill the host cells from the inside [51,52]. Escaping into the cytoplasm as well as intracellular cytotoxicity has also been shown for *S. schweitzeri* isolates [10]. In addition to the adhesins FnBPA and FnBPB, *S. schweitzeri* harbors genes for the adhesin clumping factors A and B (ClfA, ClfB), collagen-binding adhesin CNA, extracellular matrix binding protein Ebh, elastin binding protein Ebp, and the extracellular fibrinogen-binding protein Efb [10,24]. The absence of *map* (*eap*) in the genome of *S. schweitzeri* is striking. Interestingly, only fragments and not the complete *map* gene were detected in African *S. aureus* isolates [10,53].

During infection bacteria need iron which can be obtained from the blood. *S. aureus* uses a hemoglobin receptor, IsdB, to bind to hemoglobin and to steal iron-containing heme. IsdB from *S. schweitzeri* has only 77% similarity to *S. aureus*; however, it binds hemoglobin from primates with a similar pattern of species preference as *S. aureus* [54].

In summary, these results show that *S. schweitzeri* shares many virulence features with *S. aureus*.

## 7. Animal Adaption

While many *S. schweitzeri* characteristics suggest a possible transfer of the pathogen from animals to humans, the pathogen also shows characteristics that suggest an adaptation to its animal hosts. For instance, *S. schweitzeri* seems to grow better than *S. aureus* at 34 °C and 40 °C, which correspond to the respective body temperatures of bats and monkeys [10].

Adaptation to a new host such as humans would require *S. schweitzeri* to overcome the host’s immune defenses, such as T-cell-mediated immunity, the complement system, neutrophils, and phagocytes. In *S. aureus*, the acquisition and loss of mobile genetic elements (MGEs) often leads to host-specific adaptation. Sa3int phages, which are integrated into the *hlb* locus-encoding beta-hemolysin (Hlb), have a special role in this process. These phages carry the *scn*, *chp*, and *sak* genes, encoding the human-specific immune evasion factors staphylococcal complement inhibitor, and chemotaxis inhibitory protein and plasminogen activator staphylokinase, respectively [55,56]. Up to 96% of human *S. aureus* nasal isolates carry the Sa3int phages; these isolates typically have a truncated *hlb* and are thus unable to produce Hlb as visible by the absence of double hemolytic zones [55] (Figure 1). The transfer of human-adapted isolates to livestock is accompanied by the loss of Sa3int phages. Monkey-adapted *S. aureus* also lack the immune evasion factors [57]. The same was shown for *S. schweitzeri*, and the *hlb* gene was found to be intact [10]. However, *S. schweitzeri* possesses in its genome the integrase groups φ1–3, indicating the presence of the prophages [24]. Whether *S. schweitzeri* can also integrate Sa3int phages has not yet been investigated.

The bi-component leukotoxin PVL is also phage-encoded. In sub-Saharan Africa, about half of the *S. aureus* isolates from humans are PVL-positive, whereas in Germany PVL is almost absent [53]. PVL has a high species specificity, targets the human C5a receptor, and therefore attacks only human neutrophils. It has no cytolytic effect, e.g., on Java monkey neutrophils [58]. It is therefore not surprising that PVL has not been found in *S. schweitzeri* yet [10]. In contrast, a leukocidin LukAB similar to that of *S. aureus* was also detected in *S. schweitzeri*. This has a similar high cytotoxic activity with respect to human dendritic cells compared to *S. aureus*, but the effect on human neutrophils, monocytes and macrophages is reduced [59].

Many bacteria have a defense system against foreign genes, since these can also harm the bacteria. The CRISPR-Cas (clustered regularly interspaced short palindromic repeats and the CRISPR-associated genes (Cas)) system can protect bacterial cells from the integration of foreign genes in their genome. So far, little is known about the role of CRISPR-Cas in *S. aureus*, and only a few *S. aureus* seem to bear this system [60]. For *S. schweitzeri*, this has not yet been investigated.

It has not yet been shown that *S. schweitzeri* can overcome the human immune system. This could also be the reason why the pathogen has, to date, almost only been detected in animals. So far, it is still questionable whether *S. schweitzeri* can acquire genes from *S. aureus* via horizontal gene transfer that allow it to adapt to humans.

## 8. Knowledge Gaps and Outlook

Our understanding of *S. schweitzeri* as an emerging One Health pathogen is still poor. Its presence both in animals and humans with a significant imbalance towards wildlife qualifies it as a One Health pathogen. However, despite its virulence in vitro, no infections are reported yet, neither in humans nor in animals. There are still many knowledge gaps to be filled about this bacterium (Table 2). Only a comprehensive knowledge on *S. schweitzeri* allows us to predict if it might become a relevant pathogen in veterinary and human medicine. If this is the case, close contact with monkeys and bats would be a source of risk for *S. schweitzeri* infections. In addition, globalization and climate change might further facilitate the worldwide spread of this exceptionally equipped member of the *Staphylococcus* pathogen group, requiring reliable microbiologic diagnostics and clinical case ascertainment.

## Figures and Tables

**Figure 2 microorganisms-10-00770-f002:**
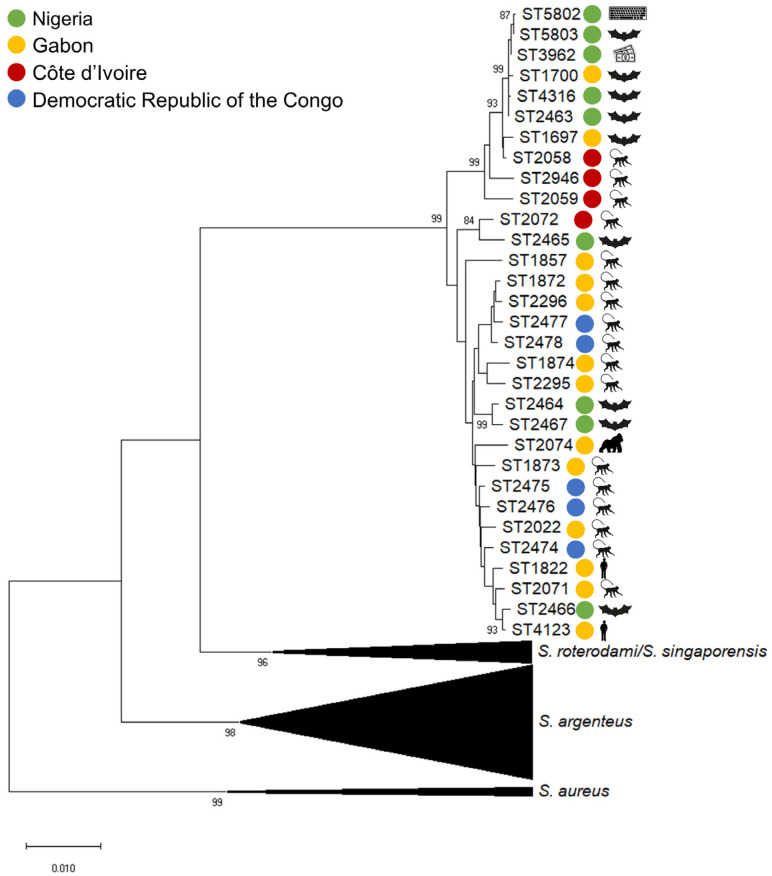
Phylogeny of *Staphylococcus schweitzeri*. The concatenated sequences of the seven alleles included in the multilocus sequence typing (MLST) scheme were used to construct a Neighbor-Joining tree for this review. The sequences were obtained from the PubMLST database (https://pubmlst.org/, accessed on 30 November 2021). The tree is rooted to the *S. aureus* sequence types (ST) ST5, ST22, ST30, ST45 and ST398 (collapsed). *S. argenteus* STs (collapsed) and *S. roterodami*/*S. singaporensis* ST were used as published elsewhere [3,5,6]. Only bootstrap values ≥ 80% were displayed.

**Figure 3 microorganisms-10-00770-f003:**
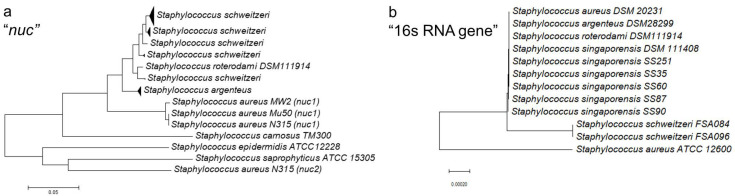
(**a**) Sequence-based identification of *Staphylococcus schweitzeri*. Neighbor-Joining trees were constructed for this review using (**a**) the complete sequences of the thermostable nuclease gene (*nuc*) and (**b**) the 16s RNA gene of members of the *S. aureus* complex. Sequences were obtained from GenBank (*nuc*: KJ748637 to KJ748639, OE998560.1, SA0746, SA1160, SAV0815, Sca_0025, SSP1444, SE100 and 16s RNA gene: P011526.1, MF678863.1, OD916897.1, MT628672, MT628673, MT628669, MT628670, MT628671, MT628674, NR_118997.2, CCEK01000031.1, CCEL01000025.1).

**Figure 4 microorganisms-10-00770-f004:**
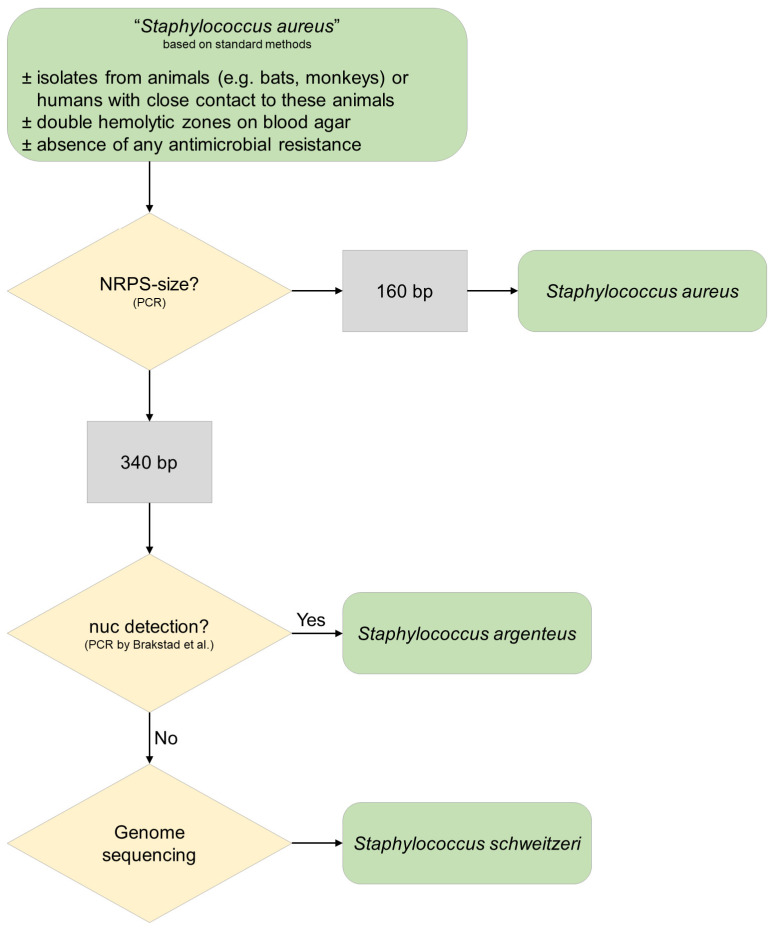
A proposed algorithm for the species identification of *Staphylococcus schweitzeri*. Colonies provisionally identified as “*Staphylococcus aureus*” should be further analyzed if they are from African wildlife and if the colonies have double hemolytic zones and do not show antimicrobial resistance. A two-step PCR approach (NRSP-PCR, *nuc*-PCR) should be followed by genome sequencing for ultimate species confirmation. There are hints that *S. roterodami* and *S. singaporensis* might be negative in the *nuc*-PCR by Brakstad et al. [37].

**Table 2 microorganisms-10-00770-t002:** Knowledge gaps and suggestions for future research on *S. schweitzeri*.

Knowledge Gap	Research Agenda
Epidemiology	To understand the geographic dispersal of *S. schweitzeri*, surveillance studies should be performed. These should include: Regions where the contact between wildlife (e.g., bats and monkeys) and humans is close; Samples from animals (wildlife and livestock) and humans; Samples from colonization and infection.
Horizontal gene transfer	The acquisition of genes by horizontal gene transfer can improve the adaptation of *S. schweitzeri* to humans and can also lead to antimicrobial resistance. It should be investigated whether gene transfer of resistance genes or virulence factors, e.g., via bacteriophages, from *S. aureus* to *S. schweitzeri* is possible.
Pathogenicity in vitro	In vitro experiments on the interaction of *S. schweitzeri* with immune cells as well as on the survival of the pathogens in whole blood (from humans, monkeys, bats) could provide information on the adaptation of *S. schweitzeri* to its natural hosts. Neutrophil transmigration under the influence of *S. schweitzeri* isolates could also be investigated to obtain information on neutrophil recruitment.
Pathogenicity in vivo	Pathogenicity should be tested in animal models to determine whether *S. schweitzeri* can cause infections. Special attention should be paid to the model; humanized mouse models, bats and monkeys appear adequate.
Capacity building	In order to gain accurate knowledge about the prevalence and distribution of *S. schweitzeri*, but also of the other members of the *S. aureus* complex, it is necessary to familiarise scientists in Africa with the subject and train them to distinguish between the staphylococci. The necessary scientific equipment for this must also be available in laboratories in Africa.
Improved diagnostics	Culture-based detection and confirmation are needed to improve the species identification of *S. schweitzeri* in resource-limited settings.

## Data Availability

Not applicable.

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
