# Peer review of "Staphylococcus schweitzeri—An Emerging One Health Pathogen?"

_microorganisms, 2022, doi:10.3390/microorganisms10040770_

Round 1

Reviewer 1 Report

Overall, I am satisfied with the authors answers, with exception of few points. 

#I would delete “Apart from Staphylococcus schweitzeri” from abstract and simply started with  Staphylococcus aureus-related complex includes xxxxx species. 

#L. 151. Important information: S. schweitzeri isolates originally identified as S. aureus are also included in this table. -> put an * specifying which ones are those in your list

The response of the authors regarding my previous  comment is still confusing

L42-45: what about the rest of the species in the complex? What is the barcoding gene for species delimitation in this group?

To the best of our knowledge, there are no barcoding genes yet available to distinguish the five species of the S. aureus related complex. Therefore, species identification of S.schweitzeri is complicated and we suggest an algorithm (Figure 4) for standardization.

# I am still bit confused about Fig 4.  Indeed, I agree that the most comprehensive approach to  delimitate among closely related species is  comparative genomics, however it is important to provide what are the main differences of these species in regard to specific discussed genes. Authors discussed  different genes such as NRPS, Nuc1, etc., as a possible candidate genes for species delimitation. It would be useful that thy put a little alignment figure, to show the main differences per gene among the discussed species.

# I also disagree with authors answer regarding of phylogenetic method.

We chose the neighbor joining tree as this method was mainly
used in similar studies (https://journals.asm.org/doi/10.1128/JCM.02327-14).

I think this is not convincing justification, since there are many studies that used other methods, for example

https://www.ncbi.nlm.nih.gov/pmc/articles/PMC3464590/

https://academic.oup.com/gbe/article/11/10/2917/5582665

https://www.sciencedirect.com/science/article/pii/S1876034116000149

https://www.frontiersin.org/articles/10.3389/fmicb.2020.02063/full

Generally the best approach is to use few methods such as NJ, ML, Bayesian and then compare. However if you have evidence that NJ for this type of study is the best accepted approach please specify and provide reference. 

Reviewer 2 Report

The authors incorporated all previous comments into the text according to the instructions and completed all the required details. For this reason, I recommend publishing the article in this form.

Author Response

The authors incorporated all previous comments into the text according to the instructions and completed all the required details. For this reason, I recommend publishing the article in this form.

Reply: We thank the reviewer for his or her kind words.

Reviewer 3 Report

The authors has accepted my suggestions so in my opinion it can be accepted in it's present form.

Author Response

Reply: We thank the reviewer for his or her kind words.

This manuscript is a resubmission of an earlier submission. The following is a list of the peer review reports and author responses from that submission.

Round 1

Reviewer 1 Report

  1. The names of bacterial species should be written in italics. Please, use italics throughout the text.
    g. lines: 37, 38
    Staphylococcus aureus, Staphylococcus argenteus, Staphylococcus roterodami, Staphylococcus schweitzeri and Staphylococcus singaporensis
  2. Lines: in vitro - Please, use a italic style (lines: 155, 171, 255).
  3. When writing gene names, please, use italics throughout the text e.g.: lines: 170, 222
    sspA and sspP, aur or sspB, scn, chp, sak genes,
  4. The authors wrote:
    ,,This review aims to summarise existing information on S. schweitzeri and to identify knowledge gaps”.
    This goal was not achieved. There is no critical assessment of the available results. The results are not adequately presented, errors appear, there is no match with the original sources.
  5. Line from 54: ,,The One Health concept recognizes that the health of humans is closely connected to the health of animals and the environment. Given the potential transmission of S. schweitzeri from animals to humans and the potential for adaptation of this bacterium in humans, it is important to understand the physiology of this potential One-Health pathogen and to set up conventional methods of identification”.

The paper does not describe the benefits of using the methods          mentioned in the manuscript (PCR, Sequencing , MALDI-TOF; which one is the most useful). Regarding mass spectrometry, which MALDI-TOF system was used and which database.

  1. In the section ,,Antibiotic resistance”, please specify which method of drug susceptibility assessment was used.
  2. line: 159
    ,,S. schweitzeri belongs to the coagulase-positive staphylococci and is able to coagulate human, rabbit, canine and equine plasma, but not porcine or avian plasma [12,19].”
    This is a very general sentence, not precise, please compare it to the study conducted by: Almut Grossmann: The majority of S. schweitzeri isolates was able to coagulate rabbit plasma (n = 57, 98%) followed by human plasma (n = 55, 95%) and chimpanzee plasma (n = 39, 67%). 39 isolates (67%) coagulated all three different plasma (Figure S1).
    In: Grossmann A, Froböse NJ, Mellmann A, Alabi AS, Schaumburg F, Niemann S. An in vitro study on Staphylococcus schweitzeri virulence. Sci Rep. 2021;11(1):1157. There is definitely no discussion with other researchers.
  3. Line 199
    ,,This phagolysosomal escape is prerequisite for the bacteria to kill the host cells from the inside, which has also been shown for S. schweitzeri”.

I think this information is also imprecise, because in the paper by Almut Grossmann: However, we detected a large variance within the different S. schweitzeri isolates (range 0.1–3.5 escaped bacteria/cells, Figure S4).

In: Grossmann A, Froböse NJ, Mellmann A, Alabi AS, Schaumburg F, Niemann S. An in vitro study on Staphylococcus schweitzeri virulence. Sci Rep. 2021;11(1):1157.

Discussion with other authors is necessary.

Reviewer 2 Report

The Staphylococcus aureus complex is pathogenic complex threatening human and animal health, and the review on this complex, especially when it includes understudied species such as S. schweitzeri is very valuable.  However, to me, this manuscript has major flows, and several details need to be carefully considered before publication.

Abstract:

This abstract is very unstructured. It is not evident for the reader what is the purpose of this review and what to expect to find in this review. What are the main questions that authors asked? Is it the pathogenicity of the S. schweitzeri?, is it geographical distribution? Or maybe reviewing the growth kinetics of this species? Or it is all these 3 aspects that can contribute to understanding the health risk caused by this species?

To me the most central point is that  S. schweitzeri can represent a risk as a pathogen for humans as it has all the virulence factors and potential to infect  human, however, this is not appearing in the abstract at all.

Introduction:

Again, this needs a better structure. It is a review, so it needs to provide complete information about the species complex, how different species are from each other within the complex, how different pathogenicity they have, how it is possible to delaminate one from another?  How different they are from each other genetically, ecologically, geographically, host-related, etc…..

What about S. simiae? As far as I know it is genetically quite similar to S. aureus.

L42-45: what about the rest of the species in the complex? What is the barcoding gene for species delamination in this group?

L44: confusing who is similar to whom and how

Table1. I would put the Source column in alphabetical order

L90: what does  “ part of core genome” means, please specify

Fig2: from where this is coming? What is the source?

Fig3, again it is very confusing how the trees were constructed. It is not clear whether it is taken from some reference or it is built in this study. If so, you need to add a small paragraph explaining how you build the trees and add a small table with the information all sequences you used ( e.g., species names, gene name, accession number, etc.)

 Neighbor joining trees, why not other methods, for example maximum likelihood? What about different phylogenetic models?

Reviewer 3 Report

This review aims to summarise existing information on S. schweitzeri and to identify knowledge gaps.

In the abstract, all text Latin names of microorganisms must be in italic. The same for in vitro.

Tables are not according to instructions for authors. This study is novelty ut 37 references for review is not enough. In table 1 is NA and in legend NB. Table 1 showing that isolated trees are not resistant. In the part of antibiotic resistance is confusing and must be re-described. 

Identification of S. schweitzeri  is described at a very poor level. I think that authors must describe more information in this part, same for pcr and sequencing.

Some conclusions are missing in study. 

In figure 1 is resource missing.

Reviewer 4 Report

The current review focused to the One Health concept recognizes that the health of humans is closely connected to the health of animals and the environment. In general the manuscript is correctly written and it sounds. The current data are novel, correctly presented and it could be of significant scientific value for scientific community. The abstract is informative and clearly presented, interesting and conclusion is logical. Overall, the study provides potentially valuable data regarding

summarise existing information on S. schweitzeri and to identify knowledge gaps.

However, I suggest that authors add several recent references in order to expand knowledge of important of mastitis in One Health approach (from human and animal aspect) and improve the manuscript.

References which I suggest in the manuscript are following:

BUROVIĆ, J. (2020): Isolation of bovine clinical mastitis bacterial pathogens and their antimicrobial susceptibility in the Zenica region in 2017. Vet. stn. 51, 47-52. (In Croatian).

BENIĆ, M., N. MAĆEŠIĆ, L. CVETNIĆ, B. HABRUN, Ž. CVETNIĆ, R. TURK, D. ĐURIČIĆ, M. LOJKIĆ, V. DOBRANIĆ, H. VALPOTIĆ, J. GRIZELJ, D. GRAČNER, J. GRBAVAC, M. SAMARDŽIJA (2018): Bovine mastitis: a persistent and evolving problem requiring novel approaches for its control - a review. Vet. arhiv 88, 535-557

LAMARI, I., N. MIMOUNE, D. KHELEF (2021): Effect of feed additive supplementation on bovine subclinical mastitis. Vet. stn. 52, 445-460.

SAIDI, R., Z. CANTEKIN, N. MIMOUNE, Y. ERGUN, H. SOLMAZ, D. KHELEF and R. KAIDI (2021): Investigation of the presence of slime production, VanA gene and antiseptic resistance genes in Staphylococci isolated from bovine mastitis in Algeria. Vet. stn. 52, 57-63.

CVETNIĆ, L., M. SAMARDŽIJA, S. DUVNJAK, B. HABRUN, M. CVETNIĆ, V. JAKI TKALEC, D. ĐURIČIĆ, M. BENIĆ (2021): Multi Locus Sequence Typing and spa Typing of Staphylococcus aureus Isolated from the Milk of Cows with Subclinical Mastitis in Croatia. Microorganisms 9, 725.

MIMOUNE, N., R. SAIDI, O. BENADJEL, D. KHELEF, R. KAIDI (2021): Alternative treatment of bovine mastitis. Vet. Stn. 52, 639-649.

KNEŽEVIĆ, K., V. DOBRANIĆ, D. ĐURIČIĆ, M. SAMARDŽIJA, M. BENIĆ, I. GETZ, M. EFENDIĆ, L. CVETNIĆ, J. ŠAVORIĆ, I. BUTKOVIĆ, M. CVETNIĆ, M. MAZIĆ, N. MAĆEŠIĆ (2021): Use of somatic cell count in the diagnosis of mastitis and its impacts on milk quality. Vet. stn. 52, 751-764. (In Croatian).